# Revisiting the exposure criterion for PTSD: Using the COVID-19 pandemic as an opportunity to assess measurement invariance of PTSD symptoms across event types

Anouk van Duinkerken[1,2]*, Peter G. van der Velden[3,4], Michel L. A. Dückers[1,2,5], Christos Baliatsas[2], Mark W. G. Bosmans[2]

**1** Faculty of Behavioural and Social Sciences, University of Groningen, Groningen, The Netherlands, **2** Netherlands Institute for Health Services Research (Nivel), Utrecht, The Netherlands, **3** Tranzo, Tilburg School of Social and Behavioral Sciences, Tilburg University, Tilburg, The Netherlands, **4** Centerdata, Tilburg, The Netherlands, **5** ARQ Centre of Expertise for the Impact of Disasters and Crises, Diemen, The Netherlands

* a.van.duinkerken@rug.nl

## Abstract

### Introduction

There is an ongoing debate about the exposure criterion of PTSD since its introduction in the DSM-III in 1980 and the global COVID-19 pandemic has fueled it. Studies examining the prevalence of PTSD following the COVID-19 pandemic are criticized because assessed stressful pandemic-related events do not fulfill the exposure criterion of the DSM-5 and ICD-11. However, if stressful pandemic-related events are indeed distinct events compared to events that fulfill the exposure criterion, measurement noninvariance of PTSD symptom clusters should occur. The aim of the present study is to test this hypothesis.

### Methods

For this purpose data from the Dutch Public Health Monitor 2022, based on a large random sample of the Dutch population (n = 72,851) was extracted. We examined the measurement invariance in the PCL-5 subscales across different categories of events that people have experienced during the pandemic, such as the sudden hospitalization of a loved one and not being able to say goodbye to a loved one who passed away due to COVID-19.

### Results

The mean PCL-5 score was 12.36. PTSD prevalence was 9.74%, higher for events fulfilling the DSM-5 exposure criterion (16.54% vs. 9.05%), with smaller difference between events fulfilling the ICD-11 exposure criterion. Model fit was acceptable

**Data availability statement:** The data underlying the results presented in the study are available from https://www.monitorgezondheid.nl/data-aanvraag (in Dutch).

**Funding:** This research was funded by the Netherlands Organization for Health Research and Development (ZonMw), grant number 10430202120002.

**Competing interests:** The authors have declared that no competing interests exist.

across all event categories. Multigroup analyses showed that constraining loadings, intercepts, and error variances did not meaningfully worsen fit, changes in CFI and RMSEA remained below thresholds, supporting measurement invariance across event types. Conclusions: The results show measurement invariance between events during the pandemic that did and that did not comply with the DSM-5 or ICD-11 exposure criterion. These results suggest that the current exposure criteria in the DSM-5 and ICD-11 may not capture all relevant events and underscore the importance of further research to guide potential updates.

## Introduction

Posttraumatic stress disorder (PTSD) is a psychological condition that emerges following exposure to one or more traumatic events [1]. In everyday life, many experiences are labeled as "traumatic," ranging from minor setbacks to stressful events. This broad and often colloquial use of psychological terminology reflects a wider cultural trend sometimes referred to as *"therapy speak,"* in which clinical concepts are increasingly used in everyday discourse [2]. While this phenomenon may increase psychological awareness, it also risks diluting the clinical meaning of concepts such as trauma, which in psychiatry denotes profound and persistent psychological suffering resulting from exposure to extreme stressors [1,3].

Posttraumatic stress disorder (PTSD) was formally introduced in the Diagnostic and Statistical Manual of Mental Disorders (DSM*)* in 1980, marking a pivotal shift in the recognition of trauma-related psychopathology. This was based largely on the psychological consequences of trauma among Vietnam War veterans and survivors of violence and disaster. Before that, similar symptom patterns had been described under different names, such as "shell shock" and "combat fatigue" [4].

Over successive editions of the DSM, the conceptualization of PTSD has evolved, particularly regarding what qualifies as a traumatic event: the exposure criterion. In the DSM Fifth Edition (DSM-5), this exposure criterion is the following [1]:

> *exposure to actual or threatened death, serious injury, or sexual violence in the following ways: direct exposure, witnessing, learning that it happened to a close family member or friend (in case of actual or threatened death, this needs to be violent or accidental), or repeated exposure through professional duties.*

In the International Classification of Diseases 11th revision (ICD-11), the exposure criterion is slightly different [3]:

> *exposure to an extremely threatening or horrific event occurring through direct experience, witnessing, or learning that it happened to a loved one (in case of the actual or threatened death of a loved one, this needs to be violent, sudden or unexpected).*

There is an ongoing debate about this exposure criterion since the introduction of PTSD in the DSM-III in 1980. The main topic of this debate is which type of events should be included and which type of events should be excluded as a potential criterion for PTSD [5]. As a result, different criteria of what constitutes a traumatic event have been formulated in the subsequent versions of PTSD in the diagnostic manuals (see Table S1 for detailed information on changes). Differentiating which events fulfill the exposure criterion is challenging, even for experts [6].

The global COVID-19 pandemic further fueled this debate [5,7–10]. Bridgland et al. [7] argue that the COVID-19 pandemic does not neatly fit the exposure criterion for PTSD, but they find that pandemic-related stressors can still elicit PTSD-like symptoms, suggesting that current diagnostic criteria may not fully capture such experiences. Norrholm et al. [9] concluded that being a close relative or friend and subsequently learning about that individual's experience (e.g., indirect exposure) with extreme anxiety/panic and fear of death during severe respiratory distress are COVID-related experiences that meet the exposure criterion of PTSD. During this pandemic, part of the population learned about the actual or threatened death of a family member or friend, directly or indirectly caused by COVID-19. Importantly, the containment measures often excluded personal contacts and visits. However, according to the DSM-5, such experiences only meet the related exposure criterion when the actual or threatened death of a family member or friend was violent or accidental. Actual or threatened deaths due to COVID-19 are not violent or accidental resulting in a situation that, similar to other potential lethal illnesses, precludes a possible PTSD DSM-5 diagnosis among the affected relatives. This distinction between violent or accidental deaths and deaths of other causes such as illness, suicide or natural disasters, seems to be arbitrary. According to the ICD-11, these events can fulfill the exposure criterion since the deaths due to COVID-19 can be relatively sudden and/or unexpected. Other stressful COVID-19-related events, such as hospitalization of a loved one, fear of infection, or being unable to say farewell due to containment measures, do not fulfill the exposure criterion in either diagnostic manual. Additionally, the media coverage of the pandemic and its adverse effects was immense but extensive media coverage of the pandemic does not fulfill the exposure criterion in either diagnostic manual, unless the exposure is work-related and involves repeated, extreme exposure to adverse details. These restrictions posed by the exposure criterion raises questions on how to evaluate PTSD symptoms arising after COVID-19 related events that do not strictly meet the exposure criterion of PTSD.

Previous non-COVID-19 empirical studies have shown that individuals exposed to potential stressful events that do not meet the exposure criterion of PTSD, nevertheless suffer from similar post-event intrusion, avoidance and negative alterations in cognitions and mood, and alternations in arousal and reactivity [11–16]. Of interest in this perspective are the studies by Brewin et al. [17] and Solomon and Canino [18]. They argue that it is unlikely that the full spectrum of PTSD symptoms is present unless there was an exposure to a very significant stressor and wonder if the exposure criterion is even necessary. Gradus and Galea [19] question the feasibility of formulating a definitive definition for the exposure criterion, since there is still an absence of an exhaustive list of specific events that fulfill the exposure criterion in the DSM-5, and also argue for possibly omitting the exposure criterion. Long ago, Breslau and Davis [20] concluded that the "*literature on disasters, civilian and wartime, and on more ordinary stressful life events does not support the view that extreme stressors form a discrete class of stressors in terms of the probability of psychiatric sequelae or the distinctive nature of subsequent psychopathology*".

Some scholars argue for maintaining or even tightening the current diagnostic criteria, emphasizing that a PTSD diagnosis cannot be based on symptoms alone [5]. Their reasoning is that PTSD is not merely a collection of symptoms but a disorder fundamentally tied to the nature of the exposure. However, this position is difficult to justify from a clinical perspective. In virtually all other psychiatric disorders, diagnosis is based on symptomatology. Excluding individuals with identical symptom profiles based solely on the nature of their exposure is rather problematic, as the comparison of the DSM-IV-TR and DSM-5 have shown [21].

As said, the COVID-19 pandemic fueled this debate about the exposure criterion. At the same time, given the large number of affected individuals, it presents a unique opportunity to empirically test whether the different symptom clusters

of PTSD - intrusion, avoidance, negative alterations in mood and cognition, and hyperarousal are distinct in individuals who experienced potentially stressful COVID-19-related events that do meet the exposure criterion compared to those who experienced events that do not meet this criterion. If these symptom clusters differ significantly between the two groups, this would indicate measurement noninvariance and suggest that PTSD symptoms manifest differently depending on whether an event meets the exposure criterion outlined in the DSM-5 and ICD-11 or not.

The aim of the present study is to test this assumption by assessing whether the traditional four-factor structure of PTSD as posed in the DSM-5 remains applicable and valid across different types of COVID-19 and non-COVID-19 related events. Additionally, it aims to assess whether traumatic events meeting the exposure criterion posed in both the DSM-5 and the ICD-11 are distinct in their factor structure compared to events that do not meet this criterion. The null hypothesis is that PTSD symptom clusters exhibit measurement invariance between events that meet the exposure criterion and those that do not. This implies that the factor structure of the two different categories is the same and thus does not provide evidence of a distinctive nature of symptoms, questioning the validity of the current exposure criterion posed in the DSM-5 and ICD-11.

## Methods

### Study design

This study is part of the Integrated Health Monitor COVID-19, a five-year, multimethod research project on the health effects of the COVID-19 pandemic in the Netherlands (for details see van Duinkerken et al. [22]). The present study is based on the Dutch Public Health Monitor Adults and Elderly monitor (DPHM) conducted in September to December 2022. The DPHM consists of a comprehensive survey encompassing various questions about health, well-being, and lifestyle, and is normally conducted every four years. During the Integral Health Monitor COVID-19, the DPHM is conducted every two years and focused on topics related to the COVID-19 pandemic.

### Procedures

The data collection for the DPHM2022 took place through a random sample drawn from the Basic Register for Population (BRP). The BRP is based on the digitized Municipal Personal Records Database and includes personal data (names, addresses, marital status) for every person that is enlisted in a municipality. The random sample that is drawn from the BRP forms a representative sample of the Dutch population. Individuals of 18 years and older were invited by letter in the fall of 2022 to complete the questionnaire digitally or on paper. Those invited to partake in the study received a letter outlining study's purpose, emphasizing the voluntary nature of participation without the need for providing a reason for refusal. The participants provided written consent. They completed the survey from September to December 2022. The net response rate of the survey was 34% (N = 364,557).

### Measures

**Exposure to potentially traumatic and stressful events.** Respondents were presented with a list of eleven potentially traumatic and other life events (from now on 'events') they could have experienced during the pandemic (from March 2020 up until the moment of responding (between September-December 2022). This list includes events specific to the COVID-19 pandemic and events that are not related to the pandemic. Table 1 presents an overview of these events and if these events fulfill the DSM-5 exposure criterion, fulfill the ICD-11 exposure criterion, or fulfills neither exposure criterion (marked by "✓").

Subsequently, when respondents indicated experiencing one or more of these events, they were asked whether they were still affected by these events. If the respondent answered 'yes,' and there were multiple events causing distress, they were asked to identify the most distressing event. Respondents were also asked to specify the time since the occurrence

**Table 1. Examined events and categorization of events.**

| Event | DSM-5 | ICD-11 | None |
|---|---|---|---|
| 1. I personally experienced hospitalization due to COVID-19 | ✓ | ✓ | |
| 2. Someone significant to me has been hospitalized with COVID-19 | | | ✓ |
| 3. Someone significant to me has passed away due to COVID-19 | | ✓ | |
| 4. I was anxious about the possibility of someone significant to me contracting COVID-19 | | | ✓ |
| 5. Within the scope of my professional responsibilities, I came into contact with individuals who experienced severe illness or passed away due to COVID-19 | ✓ | ✓ | |
| 6. Due to the implemented COVID-19 containment measures I was unable to say farewell to someone significant to me when they passed away | | | ✓ |
| 7. Someone significant to me has experienced severe illness unrelated to COVID-19 | | | ✓ |
| 8. Someone significant to me has passed away due to causes other than COVID-19 | | ✓ | |
| 9. I was subjected to threats and/or physical violence | ✓ | ✓ | |
| 10. I was subjected to sexual violence | ✓ | ✓ | |
| 11. I underwent a life-threatening accident | ✓ | ✓ | |

of this most distressing event. Some of these events may meet the ICD-11 criteria for a traumatic event. In the main analysis, only the actual death of a loved one was included, while severe illness and hospitalizations (whether related to COVID-19 or not) were excluded. Sensitivity analyses were conducted to assess whether including these events in the ICD-11 category altered the results.

**PTSD symptoms.** The respondents who experienced at least one of the 11 events, were still affected by the event and were exposed to the event more than one month ago, were administered the 20-item Dutch version of the PTSD Checklist for DSM-5 (PCL-5) [23,24]. Respondents were asked to what extent they have experienced each symptom in the past four weeks, keeping the most distressing event in mind while completing the checklist. Items of the PCL-5 are rated on 5-point Likert scales, ranging from not at all (score = 0) to extremely (score = 4). The sum score of the PCL-5 can range from 0 to 80, with higher scores indicating more severe symptoms. To determine probable PTSD, the DSM-5 diagnostic rule was applied. Following this rule, a symptom cluster was considered endorsed if the respondent rated at least one intrusion symptom (Criterion B), one avoidance symptom (Criterion C), two negative alterations in cognition and mood (Criterion D), and two alterations in arousal and reactivity (Criterion E) with a score of ≥2 (*moderately* or higher). Participants meeting all four cluster criteria were classified as having probable PTSD.

This part of the survey is included in S1 file. The final study sample consisted of 72,851 respondents that answered the PCL-5 (Cronbach's alpha = 0.948).

**Data analysis.** Data were analyzed using Stata version 18 (StataCorp LLC, College Station, TX, USA).

The mean score of the total PCL-5 and the subscales intrusion, avoidance, negative alterations in cognition and mood, and hyperarousal was calculated. Differences in mean scores across different groups were tested by using Mann-Whitney tests of difference. Comparisons were drawn between stressors that fulfill the different (DSM-5 and ICD-11) exposure criterion and those that do not. Sensitivity analyses were performed to check whether the difference between the subdomains across events can be explained by time or severity of the symptoms and this showed no explanation.

To ensure the appropriateness of the dataset for factor analysis, both Bartlett's test of sphericity and Kaiser-Meyer-Olkin's (KMO) measure of sampling adequacy were used. The results indicated favorable conditions for factor analysis, as evidenced by the significance of Bartlett's test (p = .000) and a high KMO test value of 0.9898.

Confirmatory factor analysis was used to assess the fit of the conventional DSM-5 four-factor model of PTSD across the 11 events, employing the maximum likelihood estimation method. The model was assessed independently for each event separately, and in the following categories: events that fulfill the DSM-5 exposure criterion, events that do not

fulfill the DSM-5 exposure criterion, events that fulfill ICD-11 exposure criterion and events that do not fulfill the ICD-11 exposure criterion. The Satorra-Bentler correction is used to get results that are robust for nonnormality. To systematically assess model fit, the following model fit indices were used with the following recommended cut-off criteria: comparative fit index (CFI) > 0.90, Tucker-Lewis index (TLI) >0.90 [25] and root mean square error of approximation (RMSEA) <0.06 [26].

To systematically assess measurement invariance across the different events, multi-group SEM was used to examine model fit under varying levels of constraints between groups. Changes in model fit were measured with the following changes in model fit, with their recommended cut-off criteria: $\chi^2$ difference tests, comparative fit index ($\Delta$CFI < 0.01 [27]) and RMSEA ($\Delta$RMSEA<0.015 [28]). The following combinations were compared: 1) events that fulfill DSM-5 exposure criterion versus events that do not, 2) events that fulfill ICD-11 exposure criterion versus events that do not. Sensitivity analysis were performed to check whether these models were similar for each age group.

### Ethics statement

The study was subject to ethical approval by the Medical Research Ethics Committee (MREC) of Amsterdam UMC. It was deemed exempt from further review as it was declared not subject to the WMO (Medical Research Involving Human Subjects Act). All participants have given written informed consent before they responded to the survey.

## Results

### Demographic characteristics

In total, 72,851 respondents (approximately 20% of the total sample) were exposed to and still felt affected by one of the events that they had experienced during the pandemic. This sample answered the PCL-5 and thus are included in this study. Their demographic information is included in S2 Table. A large proportion of respondents is older than 65 years old (39.9%), the sample includes relatively many women (59.6%) and about 17% of respondents had a migration background.

### PCL-5 sum score across different event categories

The mean total sum score for the PCL-5 was 12.36. When comparing the categories, those answering the PCL-5 for an event that fulfilled the DSM-5 exposure criterion had the highest score (15.82). The events fulfilling the DSM-5 exposure criterion had a higher sum score (15.82) than the events that did not fulfill the DSM-5 exposure criterion. The events fulfilling the ICD-11 exposure criterion had a higher average sum score (12.95) than those that did not fulfill the ICD-11 exposure criterion. These differences between categories were mainly explained by differences in the intrusion domain (3.97 versus 3.12, and 3.60 versus 2.85 respectively) and the avoidance domain (1.66 versus 1.28, and 1.42 versus 1.22 respectively). The exception to this was the events that fulfilled the DSM-5 exposure criterion, as they scored considerably higher on average for each domain than the events that did not fulfill the DSM-5 exposure criterion. These sum scores are presented in Table 2.

Sensitivity analyses were executed to account for possible confounders with regards to time since the event and demographics. The association between fulfilling the DSM-5 exposure criterion and the total PTSD mean score is confounded by age and financial difficulties, and that pattern is seen for all sub domains. However, the association between fulfilling the DSM-5 exposure criterion and the sum score on the PCL-5 is already small (3.81), and the change when adding age (from 3.81 to 3.17) or financial difficulties (from 3.81 to 3.16) to the model is limited. The association between fulfilling the ICD-11 exposure criterion and the sum score on the PCL-5 is confounded by age. However, the association between fulfilling the ICD-11 exposure criterion and the sum score on the PCL-5 is even smaller (1.11), and the change when adding age (from 1.11 to 1.23) is limited. More details are included in the S3 Table.

Table 2. Sum scores of the PCL-5.

| | Total, M(SE) N=72,851 | DSM-5, M(SE) N=6,674 | Not DSM-5, M(SE) N=66,177 | ICD-11, M(SE) N=33,858 | Not ICD-11, M(SE) N=38,993 |
|---|---|---|---|---|---|
| Total | 12.36 (0.05) | 15.82 (0.20) | 12.01[a](0.05) | 12.95 (0.07) | 11.85 [a] (0.07) |
| Intrusion | 3.20 (0.01) | 3.97 (0.06) | 3.12 [a] (0.01) | 3.60 (0.02) | 2.85 [a] (0.02) |
| Avoidance | 1.31 (0.01) | 1.66 (0.03) | 1.28 [a] (0.01) | 1.42 (0.01) | 1.22 [a] (0.01) |
| Negative alterations in cognition and mood | 3.90 (0.02) | 5.13 (0.08) | 3.77 [a] (0.02) | 3.97 (0.03) | 3.83 [a] (0.03) |
| Hyperarousal | 3.95 (0.02) | 5.06 (0.06) | 3.84 [a] (0.02) | 3.96 (0.02) | 3.94 [a](0.02) |

[a]Significant at the p<0.01 level compared to the sum score of the group of events that does fulfill the exposure criterion

Note. The DSM-5 and ICD-11 samples partially overlap, but they are only compared to the group that does not fulfill that exposure criterion, not to each other.

## DSM-5 prevalence across different event categories

The overall prevalence of PTSD symptoms based on DSM-5 criteria was 9.74%. This prevalence was higher among respondents who experienced an event that fulfilled the DSM-5 exposure criterion (16.54%) compared to those whose most distressing event did not meet this criterion (9.05%). A similar, though less pronounced, pattern was observed when applying the ICD-11 exposure criterion (10.37% vs. 9.19%). Events involving direct life threat or violence, such as sexual violence (61.32%), physical violence or threats (35.40%), and life-threatening accidents (17.25%), were associated with the highest prevalence. The prevalence by event is presented in Table 3. These results indicate that the average prevalence of PTSD symptoms is higher for events that meet the exposure criteria. However, even for events that do not fulfil this criterion, the symptom patterns observed still correspond to the DSM-5 symptom clusters, suggesting that such experiences can also be associated with clinically relevant posttraumatic stress reactions.

## Factor loadings of the model

Standardized factor loadings ranged from 0.54 to 0.90, indicating that each item contributed significantly to its corresponding latent factor. The factor loadings were highest in the intrusion and avoidance domains, across all groups. Detailed information on the factor loadings is included in S4 Table.

## Model fit of the different models

In Table 4, the goodness-of-fit indices are presented. The $\chi^2$ was significant in all events. There were no large variations between events with regards to model fit. In multigroup analysis, placing constraints on the factor loadings and error variances between the groups did significantly change the $\chi^2$. However, it did not considerably worsen other model fit indices. Changes in CFI and RMSEA were below its thresholds [27,28]. This implies there were no noteworthy differences between any of the group-comparisons with regard to factor structure, thus the models were invariant.

Several sensitivity analyses were performed. Firstly, the different inclusions in ICD-11 criteria did not make a difference: both models showed measurement invariance. Secondly, there were also no variations between COVID-19 related events and other events. Thirdly, running the models separately for each age group showed that measurement invariance is present across all age groups. These results can be found in S5 Table.

**Table 3. Prevalence of probable PTSD based on the PCL-5.**

| Category | Prevalence [a] |
|---|---|
| Total | 9.74 |
| Fulfills DSM-5 exposure criterion | 16.54 |
| Does not fulfill DSM 5 exposure criterion | 9.05 |
| Fulfills ICD-11 exposure criterion | 10.37 |
| Does not fulfill ICD-11 exposure criterion | 9.19 |
| Personally experienced hospitalization due to COVID-19 | 11.00 |
| Someone significant to me has been hospitalized with COVID-19 | 8.76 |
| Someone significant to me has passed away due to COVID-19 | 10.55 |
| I was anxious about the possibility of someone significant to me contracting COVID-19 | 8.83 |
| Within the scope of my professional responsibilities, I came into contact with individuals who experienced severe illness or passed away due to COVID-19 | 8.48 |
| Due to the implemented COVID-19 containment measures I was unable to say farewell to someone significant to me when they passed away | 9.94 |
| Someone significant to me has experienced severe illness unrelated to COVID-19 | 6.70 |
| Someone significant to me has passed away due to causes other than COVID-19 | 7.76 |
| I was subjected to threats and/or physical violence | 35.40 |
| I was subjected to sexual violence | 61.32 |
| I underwent a life-threatening accident | 17.25 |

[a]In percentages based on DSM-5 criteria, not on sum score.

## Discussion

### Findings

There is an ongoing debate about the exposure criterion of PTSD in the DSM-5 and ICD-11. Studies examining the prevalence of PTSD following the COVID-19 pandemic are criticized because they assessed stressful pandemic-related events which do not fulfill the exposure criterion of the DSM-5 and ICD-11. In this study, we argued that if stressful pandemic-related events are indeed distinct events compared to events that fulfill the exposure criterion, there should be measurement noninvariance of PTSD symptom clusters. The findings of this large, population-based study do not support this hypothesis. All four PTSD symptom clusters – intrusion, avoidance, negative alterations in cognition and mood, and hyperarousal – were observed across all groups, indicating measurement invariance of these clusters across the different event categories. Even though the $\chi^2$ is significant, this could be caused by the large sample size leading to incorrectly rejecting the model [29].

The analysis of PCL-5 sum scores show that the scores in the two symptom clusters most directly related to the event (intrusion and avoidance) and average prevalence is are higher for the events fulfilling the exposure criteria. It is important to note that even events that meet the current PTSD exposure criteria in the DSM-5 and ICD-11 can vary in their associated risk of developing the disorder. For instance, some events, such as rape, are linked to a significantly higher risk of PTSD compared to others, like natural disasters [30].

The combination of our results suggests two things. First, all symptom clusters can be identified across the different types of events, which is consistent with the notion that PTSD symptoms reflect the same underlying latent structure irrespective of the precipitating event. Secondly, events that meet the exposure criteria, particularly those defined by

**Table 4. Goodness-of-fit indices of the four-factor model over different events.**

| Model | χ² | p | CFI | TLI | RMSEA |
|---|---|---|---|---|---|
| Overall | | | | | |
| Total | 29800.72[a] | 0.000 [a] | 0.940 [a] | 0.931 [a] | 0.050 [a] |
| DSM-based criterion A | 3258.78 [a] | 0.000 [a] | 0.947 [a] | 0.939 [a] | 0.053 [a] |
| Not DSM-based criterion A | 26944.49 [a] | 0.000 [a] | 0.938 [a] | 0.928 [a] | 0.050 [a] |
| ICD-based criterion A | 15025.75 [a] | 0.000 [a] | 0.937 [a] | 0.927 [a] | 0.052 [a] |
| Not ICD-based criterion A | 14853.27 [a] | 0.000 [a] | 0.944 [a] | 0.935 [a] | 0.048 [a] |
| Per index event | | | | | |
| Hospitalization due to COVID-19 | 1179.41 [a] | 0.000 [a] | 0.927 [a] | 0.915 [a] | 0.060 [a] |
| Hospitalization of a loved one due to COVID-19 | 1071.35 [a] | 0.000 [a] | 0.947 [a] | 0.939 [a] | 0.047 [a] |
| Death of a loved one due to COVID-19 | 2223.56 [a] | 0.000 [a] | 0.943 [a] | 0.935 [a] | 0.051 [a] |
| Severe illness of a loved one, not COVID-19 | 4783.39 [a] | 0.000 [a] | 0.940 [a] | 0.930 [a] | 0.048 [a] |
| Death of a loved one, not COVID-19 | 9912.09 [a] | 0.000 [a] | 0.929 [a] | 0.917 [a] | 0.052 [a] |
| Fear of infection of a loved one with COVID-19 | 7282.77 [a] | 0.000 [a] | 0.946 [a] | 0.937 [a] | 0.048 [a] |
| Exposure to severely ill or dying COVID-19 patients in professional setting | 1081.60 [a] | 0.000 [a] | 0.946 [a] | 0.937 [a] | 0.044 [a] |
| Not being able to say goodbye to a loved one who passed away | 2264.57 [a] | 0.000 [a] | 0.940 [a] | 0.931 [a] | 0.048 [a] |
| (Severe threat of) physical violence | 801.82 [a] | 0.000 [a] | 0.948 [a] | 0.940 [a] | 0.058 [a] |
| Sexual violence | 353.97 [a] | 0.000 [a] | 0.925 [a] | 0.913 [a] | 0.069 [a] |
| Life threatening accident | 961.79 [a] | 0.000 [a] | 0.931 [a] | 0.920 [a] | 0.059 [a] |
| Multigroup (ICD Criterion A versus non-Criterion A) | | | | | |
| No constraints | 81761.07 | 0.000 | 0.911 | 0.898 | 0.082 |
| Same factor loadings | 83092.72 | 0.000 | 0.909 | 0.901 | 0.081 |
| Same factor loadings + intercepts | 84572.23 | 0.000 | 0.907 | 0.903 | 0.080 |
| Same factor loading + intercepts + error variances | 87686.59 | 0.000 | 0.904 | 0.905 | 0.079 |
| Multigroup (DSM Criterion A versus non-Criterion A) | | | | | |
| No constraints | 59899.69 | 0.000 | 0.934 | 0.924 | 0.070 |
| Same factor loadings | 60454.02 | 0.000 | 0.933 | 0.927 | 0.069 |
| Same factor loadings + intercepts | 61892.07 | 0.000 | 0.932 | 0.929 | 0.068 |
| Same factor loading + intercepts + error variances | 65868.40 | 0.000 | 0.927 | 0.928 | 0.068 |

[a]*Using Satorra-Bentler method.*

the DSM-5, tend to result in higher scores of PTSD symptoms. This difference is mainly in the intrusion and avoidance domain which are 'typical' PTSD symptom clusters. This difference in scores between events that meet the ICD-11 exposure criterion and those that do not is even smaller. One key difference between the ICD-11 and DSM-5 criteria is that the loss of a loved one can fulfill the ICD-11 exposure criterion but not the DSM-5 criterion.

Earlier studies showed that individuals who experienced stressful events that do not fit the PTSD exposure criteria in the subsequent DSM and ICD versions, suffer from event-related intrusions, avoidance reactions, negative alterations in cognition and mood, and hyperarousal symptoms similar to individuals who experienced stressful events that do fit the exposure criteria. Ignoring the exposure criterion, these events can lead to PTSD and in some cases, they even lead to a higher prevalence of PTSD than events that did fit the exposure criterion [11–16]. The ongoing revisions of the exposure criterion in the DSM and ICD across different versions highlight its variability and subjectivity. These ongoing revisions lead to the striking situation where individuals who experience potentially traumatic events may be diagnosed with PTSD in one version of these manuals but not in another [13,31]. This illustrates that the exposure criterion is not fixed and is open to debate and potential improvement.

From a public health perspective, it is also important to consider the broader societal impact. Even though the average prevalence is lower for events that do not fulfil the exposure criterion, these events are more common in the general population. As shown in our data, experiences such as worrying about loved ones contracting COVID-19 or being unable to say farewell due to containment measures were reported by large groups, whereas direct violence or life-threatening accidents were relatively rare. Consequently, even if only a small proportion of those affected by these widespread events develop PTSD, the absolute number of individuals suffering from clinically relevant posttraumatic distress may still be considerable.

This study suggests that reliance on a narrow exposure criterion may be overly restrictive and that the exposure criterion may warrant reconsideration. Reassessing this criterion in light of a broader spectrum of potentially traumatic experiences may facilitate a more accurate understanding and identification of PTSD across diverse populations. The observed measurement invariance indicates that trauma-related reactions reflect similar underlying symptom clusters across event types, supporting the usefulness of a dimensional approach to trauma. At the same time, further research is needed to establish whether these findings generalize to clinically diagnosed samples.

In addition, our findings support the view that COVID-19-related experiences can be meaningfully examined from a trauma perspective and that PTSD represents a relevant outcome in assessments of the pandemic's mental health impact when symptom levels exceed established cut-off values. Importantly, the current findings highlight that a thorough assessment of respondents' exposure to events remains essential. Studies must always clarify to which events the reported intrusions, avoidance reactions, negative alterations in cognition and mood, and hyperarousal symptoms relate to. It must be noted that research on PTSD resulting from the pandemic too often neglect this important detail [32], making it difficult to determine if the examined symptoms are genuinely caused by a pandemic-related event or by another event. Without identifying the triggering event, the accuracy and relevance of these studies are compromised, ultimately hindering our understanding and ability to address pandemic-induced PTSD effectively.

## Strengths and limitations

One of the key strengths of our study is the large sample size, which increases statistical power and precision of our results, making our conclusions more generalizable. Additionally, our study benefits from including various types of events, which allows for robust comparative analysis. Importantly, our sample is drawn from the general population rather than a biased group, such as individuals already seeking psychological support. This approach ensures that our findings are more reflective of the broader community and not limited to a specific subpopulation, thereby providing a more accurate representation of the experiences and behaviors of the general population.

The DPMH 2022 survey has limitations that are important to consider when interpreting the findings. One notable limitation is recall bias, where participants may not accurately remember past experiences and perceptions, potentially leading to biased responses. The time since exposure could range up to two and a half years, since the beginning of the pandemic. However, the study primarily focuses on PTSD symptoms experienced in the past four weeks, limiting the impact of potential recall bias. Another limitation is selection bias, since participation was voluntary, possibly resulting in a sample that does not fully represent the entire adult population. Despite efforts to diversify recruitment, reaching certain groups, particularly the most vulnerable, proved challenging. The sample consists of an overrepresentation of elderly, almost 40% of the sample is older than 65 years while 20.2% of the Dutch adult population was older than 65 years in 2022, and women, which form almost 60% of the sample while 50.3% of the adult Dutch population are women [33]. However, the focus of this study is on the factor structure of PTSD and not its prevalence, thus the lack of generalizability is less problematic. When looking at the age groups separately, measurement invariance is observed in all groups.

A common critique of studies that show that PTSD symptoms do arise after experiencing events that do not fulfill the exposure criterion is that they rely on self-report surveys rather than clinical interviews, which can lead to bias [3]. The same applies to to this study, all results are based on self-report measures and not on clinical diagnoses. Consequently,

this study cannot draw conclusions about clinically confirmed PTSD diagnoses. At the same time, it is important to note that PTSD cannot be measured objectively [34], both self-report and clinical interviews rely on subjective symptom reports. The PCL-5, while not a diagnostic tool, assesses the same underlying construct of PTSD symptoms and has demonstrated good correspondence with clinician-based diagnoses, including in Dutch validation studies [35]. While the current study has attempted to investigate whether PTSD symptoms arise following events that do not meet the formal exposure criterion using measurement invariance within the PCL-5, replication with clinician-administered diagnostic interviews is needed to determine whether the exposure criterion should be reconsidered in the context of clinically diagnosed PTSD.

A limitation of this study is that the data were collected during the COVID-19 pandemic, an unusual period that may have influenced participants' experiences. To enhance the generalizability of the findings, further research should examine similar traumatic events outside of the pandemic context, such as the sudden hospitalization or death of a loved one.

## Implications

The debate surrounding the PTSD exposure criterion is ongoing, and based on past revisions of the DSM and ICD, further modifications can be expected. If an exposure criterion remains a requirement for diagnosis, the critical question is how to determine which events should be included or excluded. We argue that assessing measurement invariance across different types of stressful events is a key empirical method for guiding such decisions. Events that exhibit measurement invariance with those already included in the exposure criterion could be considered for inclusion, whereas those that do not may warrant exclusion. Future research should prioritize evaluating the measurement invariance of stressful events that remain under debate, as this would provide crucial data for refining the definition of potentially traumatic events.

Importantly, our proposal to include events that demonstrate measurement invariance of PTSD symptoms does not imply that, similar to events that do meet the current stressor criteria, all events are equally severe. Only events associated with symptom levels reaching clinically meaningful thresholds, such as those captured by the PCL-5, are relevant in this context. In other words, the focus remains on trauma-related reactions that meet a clinically significant threshold, ensuring that normal stress responses in the community are not inappropriately labeled as PTSD. As events currently recognized in the exposure criterion, such as war, sexual violence, and natural disasters, differ in their psychological impact and prevalence [30], this cannot be used as an argument against including new events within the exposure criterion. This is further supported by regression analyses to show differences in prevalence rates between event types, using the same list of events and database as the current study [36]. Expanding the scope of eligible events based on measurement properties does not imply that all qualifying events have the same psychological consequences. One notable example is the DSM-5 criterion that considers learning about the actual or threatened death of a close family member or friend traumatic only if the event was violent or accidental. This framework raises questions about whether deaths from severe life-threatening infections, cancer, or suicide should also qualify. Applying measurement invariance testing to such cases could provide an objective and empirically grounded approach to determining which events should be included in the PTSD exposure criterion. This approach offers a way to move beyond subjective determinations, contributing to a more scientifically rigorous and data-driven foundation for future revisions of PTSD diagnostic criteria. In future research, this approach should be applied to diverse events, populations, and diagnostic methods (e.g., clinical interviews rather than only self-report instruments).

## Conclusion

This study provides evidence of measurement invariance between events that meet the DSM-5 and ICD-11 exposure criteria and those that do not. These findings highlight that similar PTSD symptomatology can emerge following stressful events not currently recognized as qualifying exposures, raising questions about the scope of the current exposure criteria

in both diagnostic manuals. Given this similarity, future research could explore whether extending diagnostic consideration to such events is warranted. In particular, the events assessed in this study may be relevant candidates for inclusion in the exposure criterion in clinical research settings, pending further validation.

## Supporting information

**S1 Table. Changes of the exposure criterion across subsequent versions of the DSM and ICD.**
(DOCX)

**S2 Table. Demographic information of the sample.**
(DOCX)

**S3 Table. Checking for confounding factors in the differences in mean sum score across categories.**
(DOCX)

**S4 Table. Factor loadings.**
(DOCX)

**S5 Table. Multigroup SEM for COVID-19 specific events compared to non-COVID events.**
(DOCX)

**S6 Table. Multigroup SEM per age group.**
(DOCX)

**S1 File. Questionnaire (translated to English).**
(DOCX)

## Acknowledgments

The data used in this study originates from the Dutch Public Health Monitor 2022 from the Community Health Services, Statistics Netherlands and the National Institute for Public Health and the Environment (RIVM). We would like to thank our colleagues at GGD GHOR and RIVM for providing us with the data of the Public Health Monitor 2022 and the collaboration within the project.

## Author contributions

**Conceptualization:** Anouk van Duinkerken, Peter G. van der Velden, Mark W. G. Bosmans.

**Formal analysis:** Anouk van Duinkerken.

**Funding acquisition:** Michel L. A. Dückers, Mark W. G. Bosmans.

**Methodology:** Anouk van Duinkerken, Peter G. van der Velden, Michel L. A. Dückers, Christos Baliatsas, Mark W. G. Bosmans.

**Project administration:** Mark W. G. Bosmans.

**Supervision:** Michel L. A. Dückers, Christos Baliatsas, Mark W. G. Bosmans.

**Visualization:** Anouk van Duinkerken.

**Writing – original draft:** Anouk van Duinkerken.

**Writing – review & editing:** Anouk van Duinkerken, Peter G. van der Velden, Michel L. A. Dückers, Christos Baliatsas, Mark W. G. Bosmans.

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
