## [Decision Letter · Decision Letter 0]

8 Oct 2025

Dear Dr. van Duinkerken,

Thank you for submitting your manuscript to PLOS ONE. After careful consideration, we feel that it has merit but does not fully meet PLOS ONE’s publication criteria as it currently stands. Therefore, we invite you to submit a revised version of the manuscript that addresses the points raised during the review process.

We look forward to receiving your revised manuscript.

Kind regards,

Hong Wang Fung

Academic Editor

PLOS ONE

Journal Requirements:

“This research was funded by the Netherlands Organization for Health Research and Development (ZonMw), grant number 10430202120002”

4. Please amend your authorship list in your manuscript file to include all authors name.

5. We noted in your submission details that a portion of your manuscript may have been presented or published elsewhere.  “The data that forms the basis for this study has been published in another article, however, the results are very different. This study focuses on measurement invariance and the relevance of the exposure criterion, and does not mention prevalence rates or risk factors like in the other study.” Please clarify whether this [conference proceeding or publication] was peer-reviewed and formally published. If this work was previously peer-reviewed and published, in the cover letter please provide the reason that this work does not constitute dual publication and should be included in the current manuscript.

Additional Editor Comments:

Please respond to the reviewers' comments.

Reviewers' comments:

Reviewer's Responses to Questions

**Comments to the Author**

1. Is the manuscript technically sound, and do the data support the conclusions?

Reviewer #1: Partly

Reviewer #2: Yes

Reviewer #3: Yes

2. Has the statistical analysis been performed appropriately and rigorously?

Reviewer #1: Yes

Reviewer #2: Yes

Reviewer #3: Yes

3. Have the authors made all data underlying the findings in their manuscript fully available?

Reviewer #1: Yes

Reviewer #2: Yes

Reviewer #3: Yes

4. Is the manuscript presented in an intelligible fashion and written in standard English?

Reviewer #1: Yes

Reviewer #2: Yes

Reviewer #3: Yes

Reviewer #1: Thank you for the opportunity to review this manuscript titled “Revisiting the exposure criterion for PTSD: Using the COVID-19 pandemic as an opportunity to assess measurement invariance of PTSD symptoms across event types.” I have read through the manuscript and provide my summary and comments below.

Summary

This is a very interesting manuscript that describes an analysis of a self-report measure from a large, random, and representative sample of individuals from the Netherlands who completed a general health survey in 2022. The authors looked at whether the factor structure of the post-traumatic stress symptoms screening checklist (PCL-5) differed significantly between individuals who reported COVID-19-related events meeting DSM-5 or ICD-11 exposure criterion for PTSD and those who experienced events that did not meet the DSM-5 or ICD-11 criterion. Although valuable and strengthened by the large sample size, the study conclusions are overstated and need revisions.

Comment 1:

A major limitation is that the PCL-5 is not a diagnostic test for PTSD, but rather a self-report assessment. It cannot be established, therefore, that the individuals included in the analysis were diagnosable with PTSD. Since PTSD describes a clinical entity, with criteria of duration and disturbance in functioning being essential components, the results could be interpreted as showing that the factor structure of the PCL-5 does not vary in the general population experiencing different stressful events.

Comment 2:

In addition, the finding that PCL-5 scores were significantly higher in every one of the four subscore domains of PTSD symptomatology for individuals experiencing events meeting the DSM-5 exposure criterion needs to be emphasized and could be used to argue for the dimensional nature of trauma- and stress-related disorders and reactions.

In addition, due to the significantly higher scores for events meeting the DSM-5 and ICD-11 criteria, the authors need to acknowledge the opposite conclusion which is that expanding the exposure criteria could risk expanding the diagnostic entity of PTSD to include related, milder instances of stress- or trauma-related reactions.

Comment 3:

More importantly, the fact that the results come from a population sample, rather than a clinical sample, risks mixing normal stress and trauma-related reactions with clinical ones. In my opinion, this is the most important limitation of this study.

The authors need to note that the scores of their sample were very low when compared to clinical samples of individuals with PTSD. This, again, hints at the conceptual leap necessary to conclude that the findings in this study apply beyond normal stress and trauma-related reactions. For a reference on proposed PCL-5 cutoff values, see: Pettrich, A., Schellong, J., Dyer, A., Ehring, T., Knaevelsrud, C., Krüger-Gottschalk, A., Nesterko, Y., Schäfer, I., & Glaesmer, H. (2025). Beyond one-cutoff-fits-all: determining cutoff values for the PTSD checklist for DSM-5 (PCL-5). European Journal of Psychotraumatology, 16(1), 2514878. https://doi.org/10.1080/20008066.2025.2514878

Comment 4:

In general, if the results applied to clinically diagnosed cases of PTSD, they would call into question what the DSM-5-TR states as: “Indirect exposure through learning about an event (Criterion A3) is limited to events affecting close relatives or friends that were violent or accidental (i.e., death from natural causes does not qualify). Such events include murder, violent personal assault, combat, terrorist attack, sexual violence, suicide, and serious accident or injury.” Indeed, the DSM-5-TR excluding natural deaths of loved ones, which could still be sudden and “traumatic”, does seem arbitrary. However, the evidence presented here does not clearly show this for clinical cases, and I believe, therefore, that the main conclusion of this study is overstated and needs revision.

The main findings, that “all four PTSD symptom clusters (intrusion, avoidance, negative alterations in cognition and mood, and hyperarousal) are present in all groups,” are interesting and need to be stated without confusing population-level self-reporting of symptoms with clinically distinct entities.

Comment 5:

In addition to the overstated conclusions, which are not based on clinical samples, the manuscript includes errors of language and table numbering, which further weaken the quality of the work. The entire manuscript needs careful revision of table numbering and citing in the text. Table 3 needs to be labeled as Table 2, and it is not mentioned in the text and should be cited when discussing differences in scores between events meeting DSM-5 or ICD-11 criteria. Similarly, Table 5 needs to be relabeled as Table 4.

All abbreviations need to be explained first, such as PCL-5 in the abstract. The differences between events in total PCL-5 scores and in every sub score need to be mentioned in the abstract. Statistical measures for goodness of fit similarly need to be mentioned in the abstract, or at least the relevant p-values.

Type-editing is needed for the entire manuscript to ensure accurate grammar and typing errors. For example, in the abstract, the sentence “We found The results show measurement invariance between events during the pandemic that did and that did not comply with the DSM-5 or ICD-11 exposure criterion” needs to be revised.

Thank you once again for the opportunity to review this manuscript.

Reviewer #2: This is a very good manuscript, perhaps more could have been written about the history of PTSD as a diagnosis, and the difference between the lay pub;ic understanding of trauma and the clinical diagnostic understanding.

Reviewer #3: The exposure criterion of the DSM-5 and ICD-11 is a debated issue. The authors utilized a special period - the COVID-19 pandemic, to test whether stressful pandemic-related events would result in measurement invariance of PTSD symptom clusters compared to the category of events fulfilling the exposure criterion of the DSM-5 and ICD-11.

The results demonstrated the measurement invariance between events during the pandemic that did and that did not comply with the DSM-5 or ICD-11 exposure criterion. These results challenge the current exposure criteria in the DSM-5 or ICD-11 and call for a revision of the exposure criteria.This study was well conducted and the findings have important implications.

I only have some minor points as follows:

1."n=72.851" means "n=72,851"? I think the latter is more appropriate.

2. In the "Participants" section, the number of the participants in this study and relevant information should be given.

3. In Table 1, the “x” implies "yes", but readers might misunderstand it as "no", a tick might not cause such ambiguity.

4. The implications and future study directions should not be discussed in the "Conclusions" section, which made the conclusion so long.

.

Reviewer #1: No

Reviewer #2: No

Reviewer #3: No

---

## [Author Response · Author response to Decision Letter 1]

12 Nov 2025

We thank the reviewers for their thoughtful and constructive comments on our manuscript titled “Revisiting the exposure criterion for PTSD: Using the COVID-19 pandemic as an opportunity to assess measurement invariance of PTSD symptoms across event types”. We appreciate the time and effort spent to reviewing our work, and we have carefully revised the manuscript in response to all comments. Below, we provide detailed responses.

Reviewer #1

Comment 1: A major limitation is that the PCL-5 is not a diagnostic test for PTSD, but rather a self-report assessment. It cannot be established, therefore, that the individuals included in the analysis were diagnosable with PTSD. Since PTSD describes a clinical entity, with criteria of duration and disturbance in functioning being essential components, the results could be interpreted as showing that the factor structure of the PCL-5 does not vary in the general population experiencing different stressful events.

Indeed, our study relies on self-report data rather than clinical interviews. We have clarified in the manuscript that we are not reporting clinical diagnoses, but rather assessing PTSD symptomatology as captured by the PCL-5, which is a well-validated and widely used measure.

To our knowledge, there are no measurement invariance studies using clinical diagnoses of PTSD: the operationalization of PTSD symptom clusters largely operates in questionnaire-based research. Self-report measures like the PCL-5 provide a reliable way to capture symptom variation across populations.

We have emphasized in the revised manuscript (line 345-358) that while clinical interviews are valuable, our findings are relevant for understanding how the underlying symptom structure behaves across event types in population samples. Future research could extend this work using clinical samples. However, due to current diagnostic criteria, a clinical sample of individuals exposed to certain COVID-19-related events that do not fit the current exposure criteria would be infeasible.

Comment 2: In addition, the finding that PCL-5 scores were significantly higher in every one of the four subscore domains of PTSD symptomatology for individuals experiencing events meeting the DSM-5 exposure criterion needs to be emphasized and could be used to argue for the dimensional nature of trauma- and stress-related disorders and reactions. In addition, due to the significantly higher scores for events meeting the DSM-5 and ICD-11 criteria, the authors need to acknowledge the opposite conclusion which is that expanding the exposure criteria could risk expanding the diagnostic entity of PTSD to include related, milder instances of stress- or trauma-related reactions.

We agree that reporting higher PCL-5 scores for events meeting the DSM-5 exposure criterion is important. We have added prevalence by event category based on DSM-5 diagnostic criteria, showing the proportion of respondents meeting at least one intrusion (B) symptom, one avoidance (C) symptom, two negative alterations in cognition and mood (D) symptoms, and two alterations in arousal and reactivity (E) symptoms.

As expected, prevalence is generally lower for events not meeting the exposure criterion (see table 3 in the Results section). Nonetheless, a portion of respondents in this group still meets criteria for probable PTSD, indicating that while the average severity is lower, symptomatology can be clinically relevant for affected individuals. These results highlight the dimensional nature of trauma responses without implying that all reactions to events that do not fulfil the exposure criterion are milder stress reactions.

Additionally, our proposal to include events that demonstrate measurement invariance of PTSD symptoms does not suggest that these events are identical to those currently meeting the exposure criteria in terms of PTSD symptom sum scores or prevalence. As events currently recognized in the exposure criterion, such as war, sexual violence, and natural disasters, differ in their psychological impact and prevalence (Kessler, 2017), this cannot be used as an argument against including new events within the exposure criterion. This is also described in the manuscript.

We have discussed this more explicitly in the revised manuscript, acknowledging differences in average scores while reinforcing that the measurement structure remains consistent across event types.

Comment 3: More importantly, the fact that the results come from a population sample, rather than a clinical sample, risks mixing normal stress and trauma-related reactions with clinical ones. In my opinion, this is the most important limitation of this study.

The authors need to note that the scores of their sample were very low when compared to clinical samples of individuals with PTSD. This, again, hints at the conceptual leap necessary to conclude that the findings in this study apply beyond normal stress and trauma-related reactions. For a reference on proposed PCL-5 cutoff values, see: Pettrich, A., Schellong, J., Dyer, A., Ehring, T., Knaevelsrud, C., Krüger-Gottschalk, A., Nesterko, Y., Schäfer, I., & Glaesmer, H. (2025). Beyond one-cutoff-fits-all: determining cutoff values for the PTSD checklist for DSM-5 (PCL-5). European Journal of Psychotraumatology, 16(1), 2514878. https://doi.org/10.1080/20008066.2025.2514878

We agree that scores in our population sample are lower than in clinical samples, which is consistent with expectations. A clinical sample, by definition, includes individuals seeking treatment and thus scoring higher.

At the same time, our analyses demonstrate that trauma-related symptomatology can reach DSM-5 symptom thresholds even in a general population. Importantly, the study design allows us to examine events that would not be included in clinical studies due to the strict exposure criteria (e.g., certain COVID-19-related events). This perspective offers valuable insight into the range of symptom expression, while avoiding circular reasoning about clinical versus non-clinical cases.

From a public health perspective, it is also important to consider the broader societal impact. While prevalence is generally lower among events that do not fulfil the DSM-5 or ICD-11 exposure criterion, these events often affect a larger portion of the population. As a result, the absolute number of individuals reporting probable PTSD based on the PCL-5 can still be substantial, highlighting the relevance of these findings for population-level mental health monitoring and intervention planning. As mentioned in comment 2, while low on average, a proportion of our sample did meet the criteria for probable PTSD confirming this point.

Comment 4:

In general, if the results applied to clinically diagnosed cases of PTSD, they would call into question what the DSM-5-TR states as: “Indirect exposure through learning about an event (Criterion A3) is limited to events affecting close relatives or friends that were violent or accidental (i.e., death from natural causes does not qualify). Such events include murder, violent personal assault, combat, terrorist attack, sexual violence, suicide, and serious accident or injury.” Indeed, the DSM-5-TR excluding natural deaths of loved ones, which could still be sudden and “traumatic”, does seem arbitrary. However, the evidence presented here does not clearly show this for clinical cases, and I believe, therefore, that the main conclusion of this study is overstated and needs revision. The main findings, that “all four PTSD symptom clusters (intrusion, avoidance, negative alterations in cognition and mood, and hyperarousal) are present in all groups,” are interesting and need to be stated without confusing population-level self-reporting of symptoms with clinically distinct entities.

We acknowledge that our findings do not directly apply to clinically diagnosed PTSD. We have revised the manuscript to clarify that we do not report on clinical diagnoses.

Our key finding, that all four PTSD symptom clusters are captured consistently across event types, suggests that the latent structure of PTSD symptoms is preserved even for events outside the strict DSM-5/ICD-11 exposure criterion. While we cannot generalize to clinical diagnoses, these results indicate that the strict separation of event types may not fully capture the dimensional nature of PTSD symptoms. Replication in clinical samples is added as a suggestion for future research.

Comment 5: In addition to the overstated conclusions, which are not based on clinical samples, the manuscript includes errors of language and table numbering, which further weaken the quality of the work. The entire manuscript needs careful revision of table numbering and citing in the text. Table 3 needs to be labeled as Table 2, and it is not mentioned in the text and should be cited when discussing differences in scores between events meeting DSM-5 or ICD-11 criteria. Similarly, Table 5 needs to be relabeled as Table 4.

All abbreviations need to be explained first, such as PCL-5 in the abstract. The differences between events in total PCL-5 scores and in every sub score need to be mentioned in the abstract. Statistical measures for goodness of fit similarly need to be mentioned in the abstract, or at least the relevant p-values.

Type-editing is needed for the entire manuscript to ensure accurate grammar and typing errors. For example, in the abstract, the sentence “We found The results show measurement invariance between events during the pandemic that did and that did not comply with the DSM-5 or ICD-11 exposure criterion” needs to be revised.

Thank you for pointing out these errors. All table numbering has been carefully reviewed and corrected. Abbreviations, including PCL-5, are now defined at first mention. We have also conducted thorough type-editing to address grammar and clarity throughout the manuscript.

Reviewer #2

This is a very good manuscript, perhaps more could have been written about the history of PTSD as a diagnosis, and the difference between the lay public understanding of trauma and the clinical diagnostic understanding.

We thank the reviewer for the suggestion to include a brief discussion of the history of PTSD and distinctions between public and clinical understandings of trauma. We have added relevant background in the introduction (lines 24-35) to provide context without overextending the manuscript.

Reviewer #3

Comment 1:"n=72.851" means "n=72,851"? I think the latter is more appropriate.

This was a language-related error, in Dutch we use the dots and comma’s the other way around compared to English. It has now been corrected to the proper English format.

Comment 2: In the "Participants" section, the number of the participants in this study and relevant information should be given.

The ‘participants’ section in the methods had a wrong heading. We changed it to ‘study design’. In the results section (3.1 Demographic characteristics), the number of the participants in this study and their demographic information is given.

Comment 3: In Table 1, the “x” implies "yes", but readers might misunderstand it as "no", a tick might not cause such ambiguity.

Good suggestion, we adapted it in the revised manuscript, see table 1.

Comment 4: The implications and future study directions should not be discussed in the "Conclusions" section, which made the conclusion so long.

We have now restructured the discussion section to reflect this suggestion in the revised manuscript.

---

## [Decision Letter · Decision Letter 1]

6 Feb 2026

Dear Dr. van Duinkerken,

Thank you for submitting your manuscript to PLOS ONE. After careful consideration, we feel that it has merit but does not fully meet PLOS ONE’s publication criteria as it currently stands. Therefore, we invite you to submit a revised version of the manuscript that addresses the points raised during the review process.

Our reviewers have recommended reconsideration after major revisions.

We look forward to receiving your revised manuscript.

Kind regards,

Hong Wang Fung

Academic Editor

PLOS One

Journal Requirements:

Additional Editor Comments:

Our reviewers have recommended reconsideration after major revisions.

Reviewers' comments:

Reviewer's Responses to Questions

**Comments to the Author**

Reviewer #1: (No Response)

Reviewer #3: All comments have been addressed

2. Is the manuscript technically sound, and do the data support the conclusions?

Reviewer #1: Yes

Reviewer #3: Yes

3. Has the statistical analysis been performed appropriately and rigorously?

Reviewer #1: Yes

Reviewer #3: Yes

4. Have the authors made all data underlying the findings in their manuscript fully available?

Reviewer #1: Yes

Reviewer #3: Yes

5. Is the manuscript presented in an intelligible fashion and written in standard English?

Reviewer #1: Yes

Reviewer #3: Yes

Reviewer #1: I thank the authors for their revision of the manuscript. Highlighting the weaknesses of scientific research not only shows scientific integrity, but also opens the door for future efforts to fill gaps and further expand our knowledge.

Although the authors have adequately answered my comments, the language used in the revised manuscript is far stronger in tone than their reply. As an independent reviewer, I do not think the results of this community-based, self-report, and voluntary study can be said to “seriously question” criteria sets designed for a clinical, face-to-face diagnostic process.

I believe I have explained my critique clearly in the previous round of review, and the authors expressed agreement with most points. The manuscript, including the abstract, should be revised to more explicitly convey the limitations of this approach and to acknowledge other plausible interpretations. The results should motivate more work in this area, and alternative interpretations, such as the dimensional nature of trauma-related reactions, should be acknowledged.

An example of overly strong tone in the current manuscript is the following paragraph summarizing the results:

“317 The combination of our results suggests two things. First, all symptom clusters can be identified

318 across the different (types of) events, indicating that individuals with PTSD experience the same

319 disorder irrespective of the event. Secondly, events that meet the exposure criteria, particularly

320 those defined by the DSM-5, tend to result in slightly higher scores of PTSD symptoms. This

321 difference is mainly in the intrusion and avoidance domain which are ‘typical’ PTSD symptom

322 clusters. This difference in scores between events that meet the ICD-11 exposure criterion and those

323 that do not is even smaller.”

Please note that the presence of all symptom clusters across different events does not indicate that “individuals with PTSD experience the same disorder irrespective of the event.” This is because the sample does not include individuals diagnosed with the disorder, but rather individuals from the community reporting their perceived reactions to difficult events. As a counterexample, administering a self-report measure of depression to the general public would not establish that the pattern of depressive features in the community is equivalent to clinically diagnosed major depressive disorder. This conceptual leap suggests that the manuscript does not adequately differentiate clinical from non-clinical samples. I previously commented on the marked difference in PCL-5 scores between this sample and clinical samples.

Plausible alternative interpretations of the results, which are not acknowledged, include:

* trauma-related reactions have similar underlying clusters of experiences, and further work is needed to establish this in clinically diagnosed samples

* a dimensional approach to trauma is useful, given the measurement invariance uncovered in this analysis

* focusing on symptom clusters, and expanding exposure criteria as suggested by the authors, may lead to pathologizing normal stress reactions in the community and expanding diagnostic boundaries even when PCL-5 scores are low (the opposite of the view espoused by the authors)

Thank you again for the opportunity to review this manuscript.

Reviewer #3: The authors have addressed all the concerns from the three reviewers. I have no further questions.

.

Reviewer #1: No

Reviewer #3: No

---

## [Author Response · Author response to Decision Letter 2]

10 Feb 2026

Response to Reviewer #1

We thank Reviewer #1 for their feedback. We have revised the manuscript to address the points raised, in particular by softening the language to avoid overstating conclusions based on community-based, self-report data, clarifying the distinction from clinical diagnoses, and acknowledging alternative interpretations. Below, we detail the specific changes made.

Comment: The previous version used overly strong language, including claims such as “seriously question” and “individuals with PTSD experience the same disorder irrespective of the event,” which overextended the implications of our community-based, self-report study. Alternative interpretations, including dimensional approaches to trauma, should be acknowledged, and limitations regarding clinical generalization should be emphasized.

Response: We have revised the manuscript to ensure that the language is appropriately cautious and that limitations and alternative interpretations are explicitly acknowledged. Specific changes are as follows:

1. Abstract

Original concern: The abstract implied that the results directly challenge diagnostic manuals.

Revision: These results suggest that the current exposure criteria in the DSM-5 and ICD-11 may not capture all relevant events and underscore the importance of further research to guide potential updates.

Rationale: We softened the claim to “suggest” and emphasized that further research is needed, avoiding overstatement.

2. Findings – Symptom clusters

Original concern: The manuscript overstated that all individuals with PTSD experience the same disorder across events.

Revision: First, all symptom clusters can be identified across the different types of events, which is consistent with the notion that PTSD symptoms reflect the same underlying latent structure irrespective of the precipitating event.

Rationale: We now describe structural similarity of symptoms without claiming equivalence of clinical disorder.

3. Findings – Exposure criterion

Original concern: The manuscript implied the exposure criterion is flawed without acknowledging nuances.

Revision: This study suggests that reliance on a narrow exposure criterion may be overly restrictive and that the exposure criterion may warrant reconsideration. Reassessing this criterion in light of a broader spectrum of potentially traumatic experiences may facilitate a more accurate understanding and identification of PTSD across diverse populations. The observed measurement invariance indicates that trauma-related reactions reflect similar underlying symptom clusters across event types, supporting the usefulness of a dimensional approach to trauma. At the same time, further research is needed to establish whether these findings generalize to clinically diagnosed samples.

Rationale: We now clarify the implications, highlight the dimensional perspective, and emphasize the need for replication in clinical samples.

4. Findings – COVID-19-related experiences

Revision: In addition, our findings support the view that COVID-19-related experiences can be meaningfully examined from a trauma perspective and that PTSD represents a relevant outcome in assessments of the pandemic’s mental health impact when symptom levels exceed established cut-off values.

Rationale: We clarified that only symptom levels reaching clinically meaningful thresholds are relevant, addressing the concern about pathologizing normal stress reactions.

5. Strengths & Limitations – Clinical generalization

Revision: Consequently, this study cannot draw conclusions about clinically confirmed PTSD diagnoses. At the same time, it is important to note that PTSD cannot be measured objectively [33]; both self-report and clinical interviews rely on subjective symptom reports. The PCL-5, while not a diagnostic tool, assesses the same underlying construct of PTSD symptoms and has demonstrated good correspondence with clinician-based diagnoses, including in Dutch validation studies [34]. While the current study has attempted to investigate whether PTSD symptoms arise following events that do not meet the formal exposure criterion using measurement invariance within the PCL-5, replication with clinician-administered diagnostic interviews is needed to determine whether the exposure criterion should be reconsidered in the context of clinically diagnosed PTSD.

Rationale: This explicitly acknowledges the limitation of self-report data and the need for clinical validation.

6. Implications – Avoiding pathologization

Revision: Importantly, our proposal to include events that demonstrate measurement invariance of PTSD symptoms does not imply that, similar to events that do meet the current stressor criteria, all events are equally severe. Only events associated with symptom levels reaching clinically meaningful thresholds, such as those captured by the PCL-5, are relevant in this context. In other words, the focus remains on trauma-related reactions that meet a clinically significant threshold, ensuring that normal stress responses in the community are not inappropriately labeled as PTSD.

Rationale: This addresses the reviewer’s concern regarding potential overdiagnosis and demonstrates that the clinical relevance of symptoms has been considered.

7. Conclusion – Exposure criteria

Revision: This study provides evidence of measurement invariance between events that meet the DSM-5 and ICD-11 exposure criteria and those that do not. These findings highlight that similar PTSD symptomatology can emerge following stressful events not currently recognized as qualifying exposures, raising questions about the scope of the current exposure criteria in both diagnostic manuals. Given this similarity, future research could explore whether extending diagnostic consideration to such events is warranted. In particular, the events assessed in this study may be relevant candidates for inclusion in the exposure criterion in clinical research settings, pending further validation.

Rationale: Language has been softened, the claim is grounded in measurement invariance, and we explicitly note that clinical validation is necessary before altering diagnostic criteria.

Summary:

We believe these revisions address your concerns by:

• Softening strong claims regarding clinical implications

• Clearly distinguishing community-based, self-report findings from clinically diagnosed PTSD

• Explicitly acknowledging alternative interpretations (dimensionality, structural similarity)

• Highlighting the need for replication in clinically diagnosed samples

• Ensuring that the potential expansion of exposure criteria considers only individuals showing clinically meaningful symptom levels

---

## [Decision Letter · Decision Letter 2]

18 Mar 2026

Dear Dr. van Duinkerken,

Thank you for submitting your manuscript to PLOS ONE. After careful consideration, we feel that it has merit but does not fully meet PLOS ONE’s publication criteria as it currently stands. Therefore, we invite you to submit a revised version of the manuscript that addresses the points raised during the review process.

Our reviewers have now reviewed your manuscript and recommended publication. I have some minor suggestions for your consideration: First, do you think it would make sense if you conduct a regression analysis to see which specific types of events (Criterion A or non-Criterion A event) would be particularly associated with PTSD symptoms? Second, would you consider citing the latest reviews on the prevalence of DSM-5 PTSD as well as ICD-11 PTSD and complex PTSD? Third, in the abstract, you only mentioned about measurement invariance, but I think some other findings are equally, if not more, important. Please consider providing a clearer picture in the abstract. Also, how did the handle the subjects if they endorsed more than one events?

We look forward to receiving your revised manuscript.

Kind regards,

Hong Wang Fung

Academic Editor

PLOS One

Journal Requirements:

Additional Editor Comments :

Our reviewers have now reviewed your manuscript and recommended publication. I have some minor suggestions for your consideration:

First, do you think it would make sense if you conduct a regression analysis to see which specific types of events (Criterion A or non-Criterion A event) would be particularly associated with PTSD symptoms?

Second, would you consider citing the latest reviews on the prevalence of DSM-5 PTSD as well as ICD-11 PTSD and complex PTSD?

Third, in the abstract, you only mentioned about measurement invariance, but I think some other findings are equally, if not more, important. Please consider providing a clearer picture in the abstract.

Also, how did the handle the subjects if they endorsed more than one events?

Reviewers' comments:

Reviewer's Responses to Questions

**Comments to the Author**

Reviewer #1: All comments have been addressed

Reviewer #3: All comments have been addressed

2. Is the manuscript technically sound, and do the data support the conclusions?

Reviewer #1: Yes

Reviewer #3: Yes

3. Has the statistical analysis been performed appropriately and rigorously?

Reviewer #1: Yes

Reviewer #3: Yes

4. Have the authors made all data underlying the findings in their manuscript fully available?

Reviewer #1: Yes

Reviewer #3: Yes

5. Is the manuscript presented in an intelligible fashion and written in standard English?

Reviewer #1: Yes

Reviewer #3: Yes

Reviewer #1: I have reviewed thechanges made to the latest version of the manuscript. I believe the authors have made the necessary changes to the manuscript to avoid overstated conclusons while still conveying the message behind their findings.

Thank you.

Reviewer #3: (No Response)

.

Reviewer #1: No

Reviewer #3: No

---

## [Author Response · Author response to Decision Letter 3]

23 Mar 2026

Dear Editor

We are very glad to hear that the reviewers have now reviewed our revised manuscript and recommended publication.

- Regarding the suggestion to conduct a regression analysis examining the association between specific event types (Criterion A vs. non-Criterion A) and PTSD symptoms, we agree that this is an interesting question. However, it falls outside the aim of the current study. Our focus is not on identifying which events are more likely to be associated with PTSD symptoms, but on examining whether the distinction between Criterion A and non-Criterion A events is supported. The comparison between events with regards of sum score and prevalence rates are added as context. However, we already refer to studies that show that certain events (e.g., sexual violence and physical threats or violence) are more strongly associated with probable PTSD, and have added a reference to an article of our own using the same database that supports this finding: https://doi.org/10.1016/j.jpsychires.2025.02.032

- With regard to citing reviews on DSM-5 PTSD, ICD-11 PTSD, and complex PTSD prevalence, we have chosen not to include these. The aim of this study was not to estimate prevalence, and the prevalence rates observed in our data are inherently dependent on the specific selection and number of events included. Because of this, comparisons with broader prevalence reviews would be difficult to interpret and fall outside the scope of the current manuscript.

- We have revised the abstract to provide a more balanced summary of the study, not only the key findings about measurement invariance.

- Finally, we have explained in the methods how respondents who endorsed multiple events were handled. Participants were asked to select their most distressing (“worst”) event and to report their PTSD symptoms in relation to that specific event. This ensures that symptom reports are anchored to a single index event, limiting the influence of other events.

We hope these revisions address your suggestions. Finally, we want to thank you very much, besides the reviewers, for your time and the way you handled our manuscript and revisions.

Kind regards, also on behalf of my co-authors,

Anouk van Duinkerken

---

## [Editor Report · Decision Letter 3]

1 Apr 2026

Revisiting the exposure criterion for PTSD: Using the COVID-19 pandemic as an opportunity to assess measurement invariance of PTSD symptoms across event types

PONE-D-25-16093R3

Dear Dr. van Duinkerken,

We’re pleased to inform you that your manuscript has been judged scientifically suitable for publication and will be formally accepted for publication once it meets all outstanding technical requirements.

Kind regards,

Hong Wang Fung

Academic Editor

PLOS One

Additional Editor Comments (optional):

Thank you for addressing the previous comments.
---

## [Editor Report · Acceptance letter]

PONE-D-25-16093R3

PLOS One

Dear Dr. van Duinkerken,

I'm pleased to inform you that your manuscript has been deemed suitable for publication in PLOS One. Congratulations! Your manuscript is now being handed over to our production team.

Kind regards,

on behalf of

Dr. Hong Wang Fung

Academic Editor

PLOS One